# Survival modelling and cost-effectiveness analysis of treatments for newly diagnosed metastatic hormone-sensitive prostate cancer

Michaela C. Barbier[1]*, Yuki Tomonaga[2], Dominik Menges[2], Henock G. Yebyo[2], Sarah R. Haile[2], Milo A. Puhan[2], Matthias Schwenkglenks[1,2]

**1** Institute of Pharmaceutical Medicine (ECPM), University of Basel, Basel, Switzerland, **2** Epidemiology, Biostatistics and Prevention Institute (EBPI), University of Zurich, Zurich, Switzerland

\* michaela.barbier@unibas.ch

**Data Availability Statement:** All relevant data are within the paper and its Supporting Information files.

## Abstract

### Background

In metastatic hormone-sensitive prostate cancer (mHSPC) treatment, survival benefits have been shown by adding docetaxel or recent androgen receptor axis-targeted therapies (ARATs) abiraterone, apalutamide, or enzalutamide to androgen deprivation therapy (ADT). However, the optimal treatment strategy in terms of costs and effects is unclear, not least due to high ARAT costs.

### Methods

To assess treatment cost-effectiveness, we developed a Markov cohort model with health states of progression-free disease, progressive disease and death for men with newly diagnosed mHSPC, with a 30-year time horizon. Survival data, adverse events and utilities were informed by randomized controlled trial results, our meta-analysis of re-created individual patient survival data, and publicly available sources of unit costs. We applied a Swiss healthcare payer perspective and discounted costs and effects by 3%.

### Results

We found a significant overall survival benefit for ADT+abiraterone versus ADT+docetaxel. The corresponding incremental cost-effectiveness ratio (ICER) was predicted to be EUR 39,814 per quality-adjusted life-year (QALY) gained. ADT+apalutamide and ADT+enzalutamide incurred higher costs and lower QALYs compared to ADT+abiraterone. For all ARATs, drug costs constituted the most substantial cost component. Results were stable except for a large univariable reduction in the pre-progression utility under ADT+abiraterone and very large variations in drug prices.

### Conclusions

Our model projected ADT+abiraterone to be cost-effective compared to ADT+docetaxel at a willingness-to-pay threshold of EUR 70,400/QALY (CHF 100,000 applying purchasing

**Funding:** Funding This work was part of a health technology assessment funded by the Swiss Medical Board [no grant number]. The study funders were involved in determining the research question but had no role in the study design, data collection, analysis, interpretation, decision to publish, or writing of this health economic analysis report. Disclosure MS, MP, and HY received research funding from the Swiss Medical Board via employment institution, from which MB, YT, and DM were partially funded. Unrelated to the submitted work, MS has received research funding from AbbVie, Biogen, Bristol Myers Squibb, Merck Sharpe & Dohme, Mundipharma, Novartis, and Roche via employment institution and personal fees from Bristol Myers Squibb and Sandoz. Also unrelated to the submitted work, MB has received personal fees from Vifor. All other authors have declared no conflict of interest.

**Competing interests:** The authors have declared that no competing interests exist.

power parities). Given lower estimated QALYs for ADT+apalutamide and ADT+enzalutamide compared to ADT+abiraterone, the former only became cost-effective (the preferred) treatment option(s) at substantial 75–80% (80–90%) price reductions.

## Introduction

Prostate cancer is the second most frequent cancer in men worldwide [1]. In Switzerland, it has an age-standardized incidence of 118.4 per 100,000 person-years [2] and accounts for 28.% of all new male cancer cases (15.% across sexes) [3], placing a high burden on patients and the healthcare system. It also represents the second most frequent cause of death in men (15%) [3]. Prostate cancer is characterized by a relatively slow disease progression. The onset is typically after age 50.

Patients with locally advanced and metastatic prostate cancer are initially hormone-sensitive, i.e. they respond to androgen deprivation therapy (ADT) which has previously been the mainstay therapy for these patients [4]. Most metastatic hormone-sensitive prostate cancer (mHSPC) patients will eventually develop castration-resistant prostate cancer (CRPC), an ADT-resistant form with a median survival, in the presence of metastasis, of only a few months to a few years [5–8].

In recent years, substantial advances have been made in the treatment of mHSPC by adding docetaxel chemotherapy or the novel androgen receptor axis-targeted therapies (ARATs) abiraterone, enzalutamide and apalutamide to ADT [4]. These treatments have shown relevant survival benefits compared to ADT alone [9–14]. However, they vary with regard to the occurrence of adverse effects (AEs) [15] and with regard to costs, with the originator prices of the ARATs being high. In the United States (US), the price of abiraterone has lately decreased due to the availability of a generic formulation. In many European countries including Switzerland, only branded ARATs were available until recently. In Switzerland, generic abiraterone has now received marketing approval and is reimbursed by the statutory health insurance. The aim of this analysis was to investigate the cost-effectiveness of all current systemic first-line (1L) treatments for newly diagnosed mHSPC considering most up-to-date randomized controlled trial (RCT) data including the recent TITAN survival follow-up data [9]. We also performed a budget impact analysis which is outlined as a whole in S1 Appendix.

## Materials and methods

### Model overview

We developed a 3-state Markov cohort simulation model with mutually exclusive health states of progression-free disease (PFD), progressive disease (PD) and death. We assumed a hypothetical cohort of men with a median starting age of 67 years (based on 64 to 70 years in included clinical trials) and with mHSPC newly diagnosed, either as initial diagnosis or at progression after prior local therapy. Previous systemic therapy was not allowed. We compared the following five treatment strategies: (1) ADT alone; (2) ADT+docetaxel; (3) ADT+abiraterone; (4) ADT+apalutamide; (5) ADT+enzalutamide. All mHSPC patients started in the PFD state and could either stay in this state, progress, or die (S1 Fig in S2 Appendix). Patients progressing under mHSPC treatment entered the PD state representing CRPC. In the PD state, patients would receive one or two further lines of treatment (CRPC 1L; CRPC second-line (2L)) followed by late-stage palliative care after progression, or

directly late-stage palliative care. The additional lines of treatment for CRPC were not represented in the model by separate health states but covered 'within' the PD state using tunnel states and appropriate formulae (section treatment strategies).

The model assessed costs (overall, by resource type), life years (LYs), quality-adjusted life years (QALYs) as well as incremental cost-effectiveness ratios (ICERs), expressed as costs in euros (EUR) per QALY gained. Costs and effects were accrued over a 30-year time horizon, based on a monthly cycle length with half cycle correction and a 3% discount rate. We assessed costs from a Swiss healthcare payer perspective (including direct medical care costs and follow-up treatment costs) with price year 2021. For the conversion of Swiss francs (CHF) into Euros (EUR), we applied purchasing power parities (PPP) of the Organization for Economic Co-operation and Development (0.619, based on 0.704 for Euro area 19 countries and 1.137 for Switzerland, both relative to US dollars), based on the latest available year of 2020 [16]. We applied a standard rational choice approach to rank and compare all strategies simultaneously [17]. The resulting ICERs were compared with a willingness-to-pay (WTP) threshold of EUR 70,400 (CHF 100,000) per QALY gained [16], which is sometimes tentatively considered in analyses for Switzerland [18, 19]. We performed a meta-analysis of re-created individual patient overall survival (OS) and progression-free survival (PFS) data from published RCTs to inform survival model inputs. In scenario analyses, we also used hazard ratios (HRs) of network meta-analyses of aggregate-level data and individual patient data (IPD) (see next section). Input parameter estimates related to resource use were verified with two clinical experts in oncology. We drew unit costs from publicly available Swiss sources [20–23], applied published utility values, and used a further meta-analysis of published AE rates [24]. The model was programmed in the TreeAge® software (Version Pro 2021).

## Modelling of survival and progression of disease

We assumed that patients would transition between the three health states based on transition probabilities derived from modelled OS and PFS survival curves. We did not have access to individual patient data and therefore digitalized published Kaplan-Meier (KM) OS and PFS curves (Software DigitizeIt [25]) to re-create IPD (method of Guyot et al. [26]). In the interest of homogeneity, we evaluated clinical and radiographic PFS (cPFS, rPFS) only (S1 Text in S2 Appendix). In our base case analysis, all matching RCTs reporting OS, cPFS, or rPFS curves were included (S1 Table in S2 Appendix). LATITUDE [11, 27], ENZAMET [28], and NCT02058706 [13] were excluded (for reasons given in S2 Text in S2 Appendix). We further assumed that OS results already accounted for death due to non-prostate cancer-related causes.

In order to obtain pooled OS and PFS curves and to estimate HRs and median survival from the re-created IPD across the included trials, we performed a Cox meta-analysis until the longest trial observation period available (9 years). We then extrapolated the ADT data with the best-fitting parametric distribution up to 15 years (for both OS and PFS) and obtained survival curves of the intervention strategies by applying HRs in case the proportional hazards assumption was not substantially violated. In scenario analyses, we also investigated a different parametric extrapolation, added a Gaussian frailty term (representing the different trials) to the fixed treatment effect, and applied different HRs from two network meta-analyses (NMA) by Menges et al. and by Wang et al. [15, 24]. The survival modelling was performed in R version 4.1.0. More details on the survival curve modelling are given in S3 Text in S2 Appendix.

## Treatment strategies

ADT 1L treatment consisted of either luteinizing hormone-releasing hormone agonists (90% of the patients) or the antagonist degarelix (10%), every three months, based on medical expert

feedback. Addition of intravenous (i.v.) docetaxel chemotherapy (75mg/m$^2$ body-surface area) was assumed every 3 weeks for a maximum of 6 cycles, and accompanied by prednisone (10 mg every day during 6 cycles). For ARAT-containing treatments, we assumed the following daily doses until disease progression to CRPC: abiraterone acetate 1,000 mg/day plus prednisone 5mg/day, enzalutamide 160 mg/day, and apalutamide 240 mg/day. Other concomitant medications considered during PFS (for example non-steroidal anti-androgen treatment, antiemetics, and preventive medication for osteoporosis and neutropenia) are outlined in S4 Text in S2 Appendix. We assumed a treatment adherence of 100%.

Important AEs (grades 3 and 4) leading to hospitalization were identified with the help of Swiss medical experts and included in the PFD state (S5 Text, S2 Table in S2 Appendix).

With regard to further-line treatments in the PD state, which can be very heterogeneous, we modelled CRPC 1L (abiraterone, enzalutamide, docetaxel) and CRPC 2L (docetaxel or cabazitaxel) for a certain percentage of patients (S6 Text, S3-S5 Tables in S2 Appendix).

The model also included late-stage palliative care consisting of 8Gy palliative radiotherapy, physician visits including laboratory tests, and pain medication, as well as terminal care hospitalizations.

## Costs

We considered drug costs for ADT, docetaxel, ARAT and palliative medication from the Swiss specialty list [21]. The costs of resource use for physician visits, drug administration, laboratory testing, imaging procedures (computed tomography, skeletal scintigraphy, osteodensitometry), and palliative radiotherapy were based on the Swiss tariff system for outpatient physician services (TARMED 1.09) [23] and the Swiss Analysis List [20]. We estimated inpatient AE cost from the SwissDRG 2020 system [22]. Finally, we derived one-off terminal hospitalization costs from a 2018 estimate of the Cantonal Hospital of Lucerne (S7 Text in S2 Appendix). A detailed list of all resource use assumptions and cost estimates is available in S6–S9 Tables in S2 Appendix.

## Utilities

Health state utilities for metastatic prostate cancer were derived from European and international RCT publications [29–32] as no Swiss utility values were available. For the progression-free health state, we assumed a mean utility of 0.83 for mHSPC patients on ADT [32, 33], and the same utility for patients on ADT+enzalutamide and ADT+apalutamide. We further assumed ADT+abiraterone to improve the patient utility by 0.035 compared to ADT monotherapy [30, 32, 34, 35]. Docetaxel toxicities (resulting in a utility decrement of 0.03) were considered during the 6 cycles of administration [32]. For patients in the PD state, we distinguished between health utilities during further-line treatment (CRPD 1L and 2L treatment) (0.635 [36]) and after termination of further-line treatment with terminal illness (0.40 [29]). Disutilities for AEs were also sourced from the international literature [32, 37, 38]. S10 Table in S2 Appendix details all utility and disutility parameters.

## Uncertainty analyses

To investigate the robustness of the base case results we performed deterministic and probabilistic sensitivity analyses (PSA) (S8 Text, S8–S10 Tables in S2 Appendix), and a series of scenario analyses. S11 Table in S2 Appendix outlines all scenario analyses including those assuming a loglogistic (instead of a gamma base case) extrapolation of the survival curves and large price reductions for apalutamide and enzalutamide.

## Results

### Survival results

Based on the re-created and pooled IPD, we observed a significant OS benefit for ADT+abiraterone (median OS of 76.0 months; 95% confidence interval (CI) 65.2 to 92.4) compared to ADT+docetaxel (median OS of 57.5 months; 95% CI 53.8 to 62.2) as well as a significant PFS benefit for all ARAT-containing strategies versus ADT+docetaxel (Table 1 and S12 Table in S2 Appendix). We found no OS or PFS difference between ARAT-containing strategies. Compared to ADT monotherapy, all combination strategies resulted in significantly reduced hazards of progression or mortality. Further survival analysis results are provided in Fig 1 (pooled ADT OS curves from included RCTs) and Fig 2 (OS curves of all treatment strategies), S2 Appendix (S9 Text, S2-S4 Figs, and S13 Table).

### Base case results

Over a 30-year time horizon and after discounting, the ADT strategy incurred the lowest total costs (EUR 31,524) and QALYs (3.25) per patient, followed by the ADT+docetaxel strategy (EUR 38,880; QALYs 3.97) (Table 2). All ARAT-containing strategies incurred substantially higher total costs and higher QALYs, with ADT+generic abiraterone still achieving lower costs (EUR 93,084) and QALYs (5.33) than both ADT+apalutamide (EUR 194,695; 4.90 QALYs) and ADT+enzalutamide (EUR 200,519; QALYs 5.15). In all ARAT-containing strategies, drug acquisition and drug administration costs were the most substantial cost components. They represented 89% of the overall per-strategy cost for ADT+enzalutamide, 88% for ADT+apalutamide, 75% for ADT+abiraterone, 51% for ADT+docetaxel, and 33% for ADT (S14 Table in S2 Appendix). We estimated the ICER of ADT+docetaxel versus ADT at EUR 10,205 per QALY gained. ADT+abiraterone compared to ADT+docetaxel resulted in an ICER of EUR 39,814, below the tentative WTP threshold of EUR 70,400 per QALY gained (Table 2). ADT+ apalutamide and ADT+enzalutamide were absolutely dominated by ADT+abiraterone, given higher costs and lower QALYs.

### Sensitivity analyses

In univariable sensitivity analysis, changes in the utility parameter values for the progression-free disease state, proportion of patients on CRPC 1L treatment, and treatment effect sizes (HRs) had the highest impact on the ICERs (when adjacent undominated strategies were compared). When the pre-progression utility under ADT+abiraterone treatment was reduced below the pre-progression utility under ADT+docetaxel treatment, the impact on the ICER was large and the overall conclusion for ADT+abiraterone versus ADT+docetaxel changed. The corresponding ICER exceeded EUR 70,400 (S6 Fig in S2 Appendix). When comparing ADT+docetaxel with ADT, all ICERs remained below EUR 16,000 per QALY gained and hence below the WTP threshold (S7 Fig in S2 Appendix).

**Table 1. Base case hazard ratios.**

| Endpoints | Hazard ratio (95% CI) | | | |
| --- | --- | --- | --- | --- |
| | ADT+docetaxel vs ADT | ADT+abiraterone vs ADT | ADT+apalutamide vs ADT | ADT+enzalutamide vs ADT |
| OS | 0.81 (0.73, 0.90) | 0.60 (0.51, 0.72) | 0.64 (0.54, 0.75) | 0.59 (0.43, 0.81) |
| PFS | 0.71 (0.64, 0.79) | 0.45 (0.38, 0.54)[a] | 0.49 (0.41, 0.58) | 0.46 (0.37, 0.57) |

ADT, androgen deprivation therapy; CI, confidence interval; OS, overall survival; PFS, progression-free survival; vs, versus.

[a] from network meta-analysis of aggregate-level data (S13 Table in S2 Appendix).

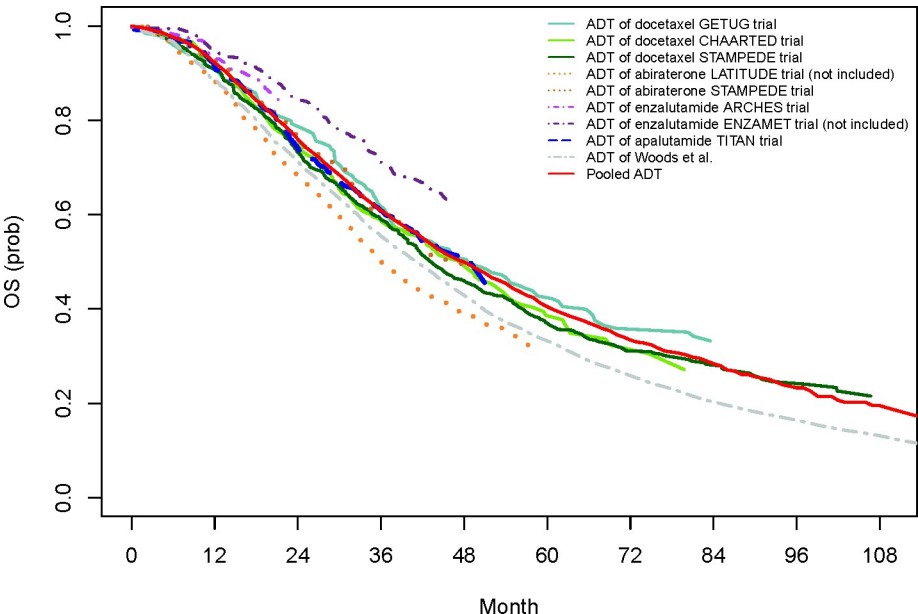

**Fig 1. Pooled OS curve for ADT and re-created Kaplan Meier OS estimates for the ADT arms of each trial.** Abi, abiraterone; ADT, androgen deprivation therapy; Apa, apalutamide; Doc, docetaxel; Enza, enzalutamide; OS, overall survival. Comment: ENZAMET and LATITUDE were not included in the meta-analysis.

PSA showed that ADT+enzalutamide and ADT+apalutamide generated substantially higher costs for moderate QALY increases compared to ADT+abiraterone, with a large overlap in their ICER clouds (Fig 3). For the WTP threshold of EUR 70,400 per QALY gained, ADT+abiraterone had the highest probability of being cost-effective, of 83.2%, followed by ADT+docetaxel with a probability of 16.5% (S8 Fig in S2 Appendix).

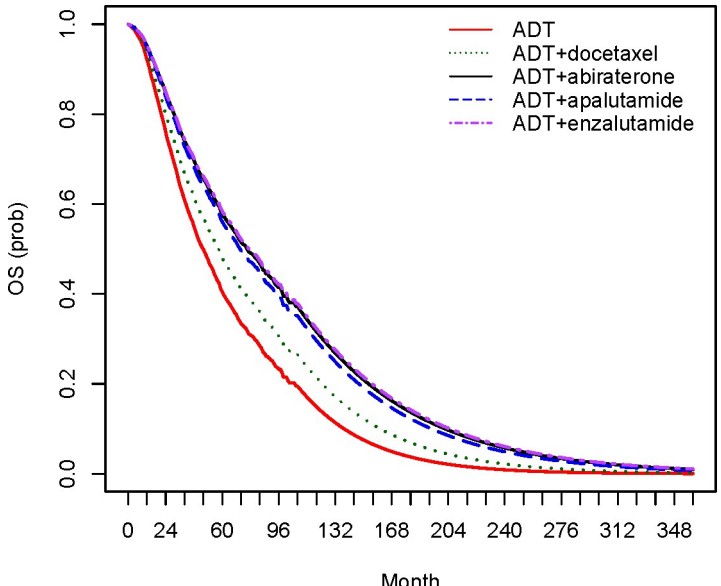

**Fig 2. Estimated OS curves for ADT and intervention strategies (gamma extrapolation).** ADT, androgen deprivation therapy; OS, overall survival.

**Table 2. Results of the base case (per person).**

| Treatment | Cost[a] (EUR) | QALY[a] (LY[a]) | Comparison to previous undominated strategy | | | | |
|---|---|---|---|---|---|---|---|
| | | | Δ Cost (EUR) | Δ QALY | Δ LY | Pairwise ICER (EUR) | Comment |
| ADT | 31,524 | 3.25 (4.77) | | | | | |
| ADT + docetaxel | 38,880 | 3.97 (5.53) | 7,357 | 0.72 | | 10,205 | |
| ADT + abiraterone | 93,084 | 5.33 (6.75) | 54,204 | 1.36 | | 39,814 | |
| ADT + apalutamide | 194,695 | 4.90 (6.50) | | | | | Absolutely dominated |
| ADT + enzalutamide | 200,519 | 5.15 (6.79) | | | | | Absolutely dominated |

ADT, androgen deprivation therapy; EUR, euros; ICER, incremental cost-effectiveness ratio; LY, life year; QALY, quality-adjusted life year.

[a] discounted.

## Scenario analyses

When using the different HRs from the NMA by Menges et al. (including or excluding data of the ENZAMET and LATITUDE trials) (S13 Table in S2 Appendix), the ICER for ADT+abiraterone versus ADT+docetaxel increased to above EUR 50,000 per QALY gained, but stayed below the assumed WTP threshold (S15 Table in S2 Appendix). In addition, inclusion of

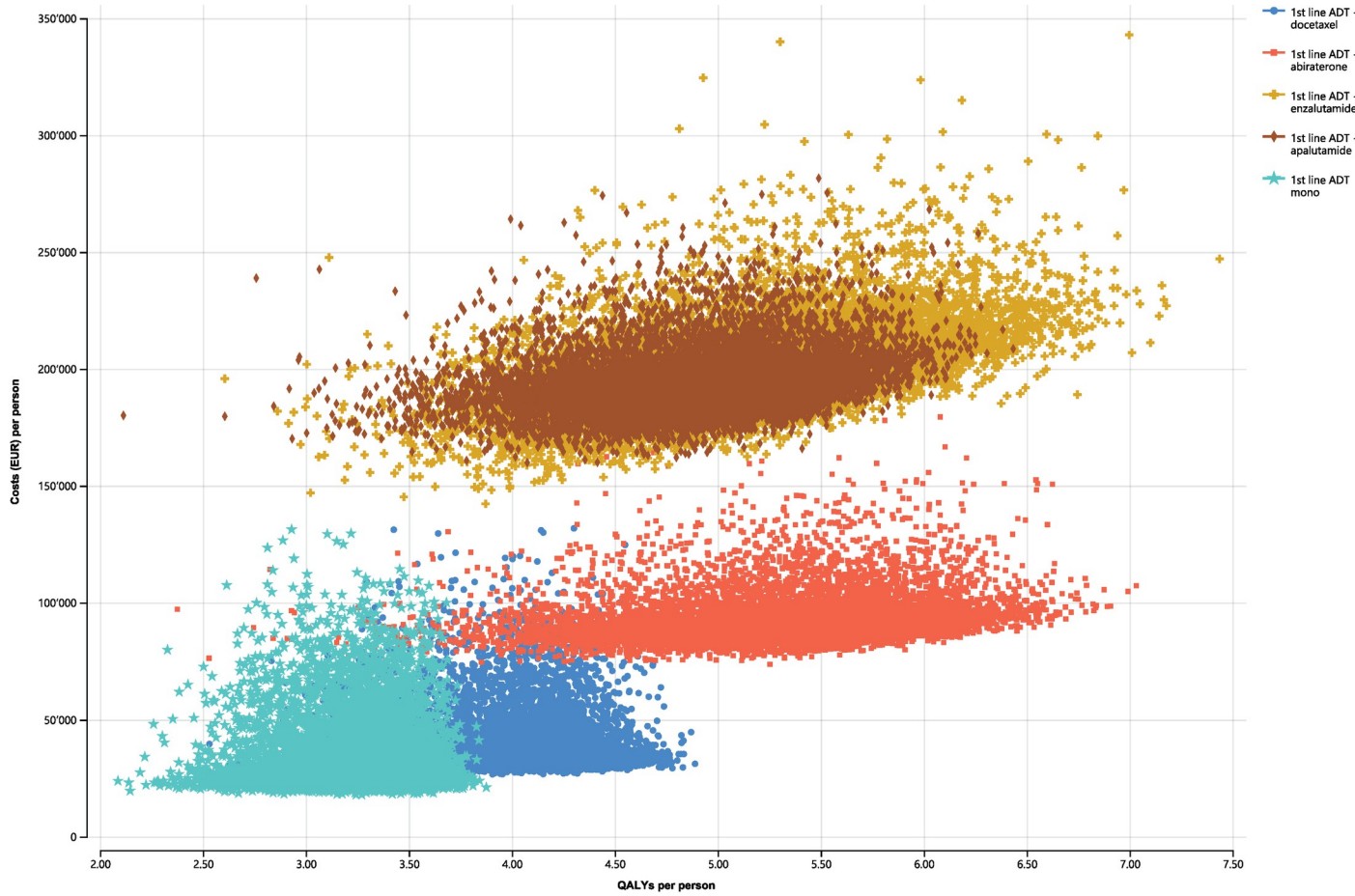

**Fig 3. Cost-effectiveness scatter plot based on probabilistic sensitivity analysis.** ADT, androgen deprivation therapy; EUR, euros; QALYs, quality-adjusted life years.

ENZAMET changed the ICER of ADT+enzalutamide compared to ADT+abiraterone from absolutely dominated to an ICER of EUR 456,383, but did not impact the overall conclusion.

An apalutamide price reduction by 80% resulted in an ICER of EUR 38,150/QALY for ADT+apalutamide versus ADT+docetaxel, while ADT+abiraterone was still cost-effective versus ADT+apalutamide (ICER of EUR 43,424/QALY) (S16 Table in S2 Appendix). At a 90% price reduction, ADT+apalutamide was the preferred treatment strategy.

Similarly, an enzalutamide price reduction by 75% resulted in an ICER of EUR 39,205/QALY for ADT+enzalutamide versus ADT+docetaxel, and of EUR 51,199 for ADT+abiraterone versus ADT+enzalutamide (S17 Table in S2 Appendix). At an 80% price reduction, ADT+enzalutamide was the preferred treatment strategy.

Assumption of the originator abiraterone price led to substantially higher costs for ADT+abiraterone, with the ICER of ADT+abiraterone versus ADT+docetaxel now being EUR 115,074 per QALY gained (S15 Table in S2 Appendix). In this scenario, ADT+docetaxel was the only cost-effective treatment strategy.

The more the time horizon was shortened, the lower the costs and QALYs, and the higher the ICERs became, but with no impact on the overall conclusion. Noteworthy, a time horizon of only 15 years resulted in ICERs very similar to the base case ICERs, meaning that follow-up years beyond 15 years and up to 30 years did not significantly affect the results.

All other investigated scenarios exerted minor to moderate influences on costs, QALYs and the ICER results, with no change in the overall conclusions.

## Discussion

We simultaneously assessed the cost-effectiveness of all five approved mHSPC drugs in a European setting with most up-to-date clinical data and considering the recent availability of abiraterone at a generic price. We found ADT+abiraterone to be the preferred treatment option at an assumed WTP threshold of EUR 70,400 per QALY gained.

Our results are in line with a cost-effectiveness analyses (CEA) by Sung et al. [33] from a US payer perspective (not yet integrating the year 2020 TITAN OS update [9]) which suggested that ADT+abiraterone is the preferred mHSPC treatment strategy in the US.

Our finding of ADT+docetaxel being cost-effective versus ADT is also consistent with previous CEAs which compared either fewer strategies simultaneously or based on earlier data, and from the perspectives of North America [32, 39, 40], China [41–43], Hong Kong [44], the UK [45], and France [46]. Compared with ADT monotherapy only and from a Canadian perspective, ADT+apalutamide treatment was unlikely to be cost-effective versus ADT [47].

Our approach is innovative in that it evaluated all currently available treatments for mHSPC simultaneously in a European setting including more mature clinical data compared to earlier analyses and the new abiraterone generic price in the base case. In addition, our model investigated the impact of the new abiraterone price on the possibility of apalutamide and enzalutamide to become a cost-effective or even the preferred treatment options. Further strengths are a refined survival analysis (meta-analysis of recreated and most up-to-date individual survival data of all current mHSPC treatments, comparison of different survival models and curve extrapolations), a homogenous PFS definition (cPFS, rPFS) and that the model parameters were based on a full-scale, systematic health economic literature review.

The analysis also has several limitations. First, meta-analysis was necessary due to scarcity of head-to-head trials directly comparing more than two treatment strategies. Second, available OS data were immature for some intervention strategies, so that sizeable extrapolations were necessary to estimate costs and QALYs over the 30-year time horizon (ADT+enzalutamide: median follow-up time of 14.4 months; ADT+apalutamide: 22.7 months for PFS). Any results

and conclusions drawn from our model for enzalutamide treatment, but also for the other treatments, thus need to be treated with caution, until longer-term data is available.

Third, because of the lack of cPFS and rPFS curves from matching RCTs for ADT+abiraterone to re-create IPD, we used one HR for PFS (for ADT+abiraterone versus ADT) from a NMA of aggregate-level data [24]. Scenario analyses based on other HRs and inclusion of data from LATITUDE, ENZAMET, and NCT02058706 did however not affect the overall conclusion of the analysis.

A further limitation is the fact that some included trials only reported AE results at a low level of detail (not always by AE type or only grades 3–5). This may have led to a certain degree of bias present in the modelled AE rates. Also, utility decrements due to AE were incorporated into the model in a simplified manner: They were combined with estimated durations and included in an aggregated way in the first model cycle. We consider the AE-related elements in our model to provide an approximation only but regard a major impact on the ICER results as unlikely.

There is also uncertainty about the types and the order of further treatment lines currently administered to patients with metastatic CRPC in Switzerland, and about the proportion of patients receiving these. While we did not investigate alternative scenarios assuming different combinations of further treatment lines, scenario analyses showed that variation in the proportion of patients receiving further treatment lines influenced the results. However, the overall conclusions regarding the cost-effectiveness of the compared treatment strategies remained the same.

## Conclusion

Our meta-analysis showed a survival benefit of ADT+abiraterone compared to ADT+docetaxel. At its generic price, ADT+abiraterone was a cost-effective treatment option versus ADT+docetaxel, at an assumed WTP threshold of EUR 70,400, and may be considered as a standard treatment option.

ADT+apalutamide and ADT+enzalutamide were not cost-effective at their originator prices; they incurred higher costs and lower QALYs than ADT+abiraterone. Based on most current clinical data and the underlying model assumptions, 75–80% price reductions would be necessary for them to become cost-effective versus ADT+docetaxel, whereas 80–90% price reductions would be necessary for them to become the preferred treatment options. Longer follow-up data and further studies of enzalutamide and apalutamide are necessary to judge their clinical value and cost-effectiveness with more accuracy. While cost-effectiveness results are not directly transferable between countries, we would expect a similar situation in other European countries with similar price structures and medical practice patterns.

## Supporting information

**S1 Appendix. Budget impact analysis.**
(DOCX)

**S2 Appendix. Cost-effectiveness analysis.**
(DOCX)

## Acknowledgments

The authors thank PD Dr. med. Richard Cathomas (Medical Oncology, Cantonal Hospital Chur, Switzerland), PD Dr. med. Christian Rothermundt (Medical Oncology, Cantonal Hospital St. Gallen, Switzerland) and PD Dr. med. Cédric Panje (Radiation Oncology, Cantonal Hospital St. Gallen, Switzerland) for their medical expert advice.

## Author Contributions

**Conceptualization:** Michaela C. Barbier, Yuki Tomonaga, Dominik Menges, Henock G. Yebyo, Sarah R. Haile, Milo A. Puhan, Matthias Schwenkglenks.

**Formal analysis:** Michaela C. Barbier, Yuki Tomonaga.

**Methodology:** Michaela C. Barbier, Matthias Schwenkglenks.

**Supervision:** Matthias Schwenkglenks.

**Writing – original draft:** Michaela C. Barbier, Yuki Tomonaga.

**Writing – review & editing:** Dominik Menges, Henock G. Yebyo, Sarah R. Haile, Milo A. Puhan, Matthias Schwenkglenks.

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
