## [Decision Letter · Decision Letter 0]

12 Oct 2022

PONE-D-22-24431Survival modelling and cost-effectiveness analysis of treatments for newly diagnosed metastatic hormone-sensitive prostate cancerPLOS ONE

Dear Dr. Barbier,

Thank you for submitting your manuscript to PLOS ONE. After careful consideration, we feel that it has merit but does not fully meet PLOS ONE’s publication criteria as it currently stands. Therefore, we invite you to submit a revised version of the manuscript that addresses the points raised during the review process. These are all relatively minor.

We look forward to receiving your revised manuscript.

Kind regards,

Christopher J.D. Wallis, MD, PhD

Academic Editor

PLOS ONE

Journal Requirements:

Reviewers' comments:

Reviewer's Responses to Questions

**Comments to the Author**

1. Is the manuscript technically sound, and do the data support the conclusions?

Reviewer #1: Yes

Reviewer #2: Yes

2. Has the statistical analysis been performed appropriately and rigorously? 

Reviewer #1: Yes

Reviewer #2: Yes

3. Have the authors made all data underlying the findings in their manuscript fully available?

Reviewer #1: Yes

Reviewer #2: Yes

4. Is the manuscript presented in an intelligible fashion and written in standard English?

Reviewer #1: Yes

Reviewer #2: Yes

5. Review Comments to the Author

Reviewer #1: Thank you for the opportunity to review the manuscript entitled, “Survival modelling and cost-effectiveness analysis of treatments for newly diagnosed metastatic hormone-sensitive prostate cancer”. Overall, this is an extremely thorough and well-structured model. The appendix is extensive and demonstrates the considerable thought and effort that was invested in this project. The BIA, while the numbers themselves not being specifically relevant to the majority of readers, highlights the overall impact of these agents on funding decisions within a healthcare budget. There are a few questions that this paper raised which are outlined below:

Questions:

• What do you mean by re-creating individual patient OS and PFS data from RCTs? Simply clarify if access to individual patient level data was obtained because this component of the methods is slightly unclear.

• What is the rationale for 1 month cycle lengths?

• Further elaboration and presentation of the supporting evidence for the choice of a higher utility for the ADT+abiraterone strategy would be worthwhile as the conclusion does change with variation of this one variable

Copy Editing Comments:

• Reference 32 and 34 are the same paper

• Line 356-357 is missing a word

Reviewer #2: Thank you for submitting your manuscript entitled, “Survival modelling and cost-effectiveness analysis of treatments for newly diagnosed metastatic hormone-sensitive prostate cancer.” In this manuscript, the authors perform a CEA assessing ARAT and docetaxel added to ADT therapy strategies in mHSPC.

1. Overall, the authors have performed a detailed CEA analysis and the methodology appears to be robust. The most important clinical (1L, 2L, palliative, AE) and methodological considerations appear to be well captured.

2. Novelty: The largest criticism is relative to the impact of the manuscript. While the methodology itself appears well done, as the authors note in their Discussion, multiple prior CEAs on this topic have been previously reported. In addition, the prior CEAs reach overall similar conclusions to the one presented here, though the authors are able to report increased follow-up and better refine the inputs. Furthermore, with the similar estimated survival curves (Fig 2), it is not surprising that abiraterone is shown to be the most cost-effective ARAT strategy given the cost benefit and the modelled utility benefit.

3. The other most significant limitation is the duration of the extrapolation relative to the modelling horizon. This is touched upon in the Discussion for enzalutamide primarily, but remains an important challenge for all arms.

4. (Minor) More details regarding AE should be included – how they were modelled – incidence, could patients have multiple, etc.? They are only briefly covered in the A5 Supplement.

5. (Minor) Can the authors provide the tunnels/formulae construction for 1L vs 2L vs palliative for CRPC given that they were not separate health states.

6. PLOS authors have the option to publish the peer review history of their article (what does this mean?). If published, this will include your full peer review and any attached files.

Reviewer #1: No

Reviewer #2: No

---

## [Author Response · Author response to Decision Letter 0]

21 Oct 2022

Please see document Response to Reviewers

---

## [Editor Report · Decision Letter 1]

25 Oct 2022

Survival modelling and cost-effectiveness analysis of treatments for newly diagnosed metastatic hormone-sensitive prostate cancer

PONE-D-22-24431R1

Dear Dr. Barbier,

We’re pleased to inform you that your manuscript has been judged scientifically suitable for publication and will be formally accepted for publication once it meets all outstanding technical requirements.

Kind regards,

Christopher J.D. Wallis, MD, PhD

Academic Editor

PLOS ONE
---

## [Editor Report · Acceptance letter]

27 Oct 2022

PONE-D-22-24431R1 

Survival modelling and cost-effectiveness analysis of treatments for newly diagnosed metastatic hormone-sensitive prostate cancer 

Dear Dr. Barbier:

I'm pleased to inform you that your manuscript has been deemed suitable for publication in PLOS ONE. Congratulations! Your manuscript is now with our production department. 

Kind regards, 

on behalf of

Dr. Christopher J.D. Wallis 

Academic Editor

PLOS ONE